

# The impact of roads and sediment basins on simulated river discharge and sediment flux in an experimental catchment designed to improve ecosystem services

**S. I. Saad[1], H. R. da Rocha[2], and J. Mota da Silva[2]**

[1]Graduate Program of Environmental Science, University of Sao Paulo, Sao Paulo, Brazil
[2]Department of Atmospheric Science, University of Sao Paulo, Sao Paulo, Brazil

Received: 11 November 2015 – Accepted: 2 December 2015 – Published: 19 January 2016

Correspondence to: S. I. Saad (sandraisaad@gmail.com)

Published by Copernicus Publications on behalf of the European Geosciences Union.



## HESSD

doi:10.5194/hess-2015-490

**Effect of roads in InVEST sediment and streamflow simulation**

S. I. Saad et al.

## Abstract

Payment of Environmental Services (PES) projects have grown our perception on the dependency of nature, yet the quantification and valuation of Environmental Services (ES) are key to evidence-base management. The modelling of the effect of deforestation in the water and sediment budgets of rural catchments usually prescribes baseline scenarios with fully pristine vegetation. Such comparison however is hardly associated with the landscape conditions where observational data is collected to provide model calibration. For example, the existence of unpaved roads and small water retention basins (containments) are potential controls of runoff and erosion in small catchments. This work shows the impact of roads and *barraginhas* (small sediment retention basins nearby the roads) on the water and sediment fluxes in a 12 km$^2$ catchment area in Extrema city, Brazil, which provides water for the Cantareira System's water reservoirs that supply about 50 % for the Sao Paulo megacity, and enrolled in the Water Producer Program (Water National Agency) as the first Brazilian PES project. Simulations with the InVEST model using high spatial resolution (5 m × 5 m) included the description of unpaved roads and *barraginhas*. Calibration used river discharge and sediment flux estimated from water turbidity measurements. Sediment concentration was estimated both with the observation and simulations, and annual comparisons seemed reasonable for mean annual estimates. Unpaved roads produced sediment export 5 times higher compared to a scenario with no roads, and potentiated the effect of *barraginhas* on sediment reduction. This study showed the benefit from understanding effects of representation of the landscape particularities in modelling such as the roads, which apart from affecting calibration, are important issue for providing efficient modelling of the effect of the Best Management Practices in the landscape scale. We also demonstrated usefulness of our research methodology and its possible applications on simulations of sediment concentration and streamflow in InVEST with few samples of observed data and thus quantify the impacts of land use change on hydrology in any other watershed.

**HESSD**

doi:10.5194/hess-2015-490

**Effect of roads in InVEST sediment and streamflow simulation**

S. I. Saad et al.

# 1 Introduction

The growing perception on the benefits human gain from nature has driven attention to the Environmental Services (ES) evaluation (MEA, 2003) and encouraged the onset of Payments for Environmental Services (PES) experiences, especially in Latin America (Martin-Ortega et al., 2012). In Brazil, the Conservador das Águas project (Water Conservation Project) was the first PES municipal experience in the Country, located in the municipality of Extrema, in the State of Minas Gerais, motivated by the known relationship between forests and water conservation, and the growing anthropic pressure around Cantareira system's water reservoirs that supply about 50 % for the Sao Paulo megacity. The project is part of the Water Producer Program, a national program of The National Water Agency of Brazil, and it was initiated with the creation of the Municipal Law 2100/2005, which provides funding to farmers that contribute to the provision of hydrological environmental services. Like all other PES experiences in Latin America, payments are not determined by the results of the action on the ES (Martin-Ortega et al., 2012). The payments are based on opportunity cost considering the area of the entire property, for owners willing to execute Best Management Practices (BMP), including reforestation of riparian zones areas especially around the springs. Like Costa Rica PES experience, used as benchmark around the world (Pagiola, 2008), a better understanding of the hydrological benefits of the actions of Conservador das Águas is needed to attract investment (Richards et al., 2015).

In order to boost development of the science of ES, InVEST (Integrated Valuation of Environmental Services and Tradeoffs) model was created aiming at helping to include the ES in decision making (Dai, 2009). It is a tool that can help users to map ES and to understand the impact of land use change. Although it does not require by the users a profound knowledge in all the approached subjects, without an adequate calibration, it can only focus a first-order assessment, rather than an accurate prediction (Hamel et al., 2015). While the data needed for parameters definition and validation

processes are sometimes difficult to obtain, the lack of comparison with observed data and inadequate validation jeopardize its use (Hamel and Guswa, 2015).

Posses watershed was the first to be part of Conservador das Águas project, with the lowest forest conservation levels at the time of its implementation (Kfouri and Favero, 2011). Since the begging of the project, some actions were performed in order to increase vegetation cover with native species and implement best management practices such as *barraginhas*, small sediment retention basins nearby the roads. For environmental monitoring proposes, the $12\,km^2$ watershed received 5 pluviometers and two rules for river level acquisition, as well as some samples of water quality, both provided by The National Water Agency. Unfortunately, the data acquisition was initiated just in 2009, after some land use modification has already been taken, turning observational studies of water quality or streamflow change more difficult. Other difficulty includes few observational data, especially from water quality, needed even for simulation studies. Apart from it and from the land change particularities of Posses, the watershed has 115 km of unpaved roads, which can play an important role on sediment transport and water quality matters (Luce, 2002). On the other hand, most modelling studies of sediment delivery and streamflow did not considered the presence of roads in their land use maps or they did not approach the issue of their effect on hydraulic connectivity, for changing the original path of water (e.g. Bangash et al., 2013; Hamel and Guswa, 2015; San, 2015; Strauch et al., 2013; Terrado et al., 2014).

The objective of this work was to adjust InVEST model for streamflow and sediment delivery simulations, in the particular conditions of Posses watershed. Representing the first watershed in Brazil to be contemplated to a PES project guaranteed by a municipal law, it has passed through land use change pattern for improvement of environmental quality, such as the implementation of *barraginhas*, avails few observational data, especially of water quality, and its crossed by unpaved roads, which not only presents higher soil loss in comparison to other land use, but also modify water and sediment path.

Discussion Paper | Discussion Paper | Discussion Paper | Discussion Paper |

**HESSD**

doi:10.5194/hess-2015-490

**Effect of roads in InVEST sediment and streamflow simulation**

S. I. Saad et al.

## 2   Methodology

### 2.1   Study area

The study area is Posses River Basin, a small basin of 12 km$^2$ located in Extrema city, in the state of Minas Gerais (Fig. 1), the first basin to be part of Conservador das Águas project. Since the beginning of the project, some efforts were made in order to improve environmental quality such as: fencing off remaining native forest areas, reforestation of native species (Richards et al., 2015), and construction of Barraginhas, small sediment retention basins nearby the roads (Fig. 2). Pastureland is the main land use in the basin (Fig. 3), as a consequence of the extensive subsistence farming, the main economic activity.

Posses has a tropical highland climate, with altitudes between 952 m (in the mouth) and 1452 m (in the head) (Fig. 4). Average temperature varies between 14.5 °C in the winter and 21.5 °C in the summer, with the rainy season in the warmer months (Fig. 5). Daily temperature amplitudes reach its maximum in the end of winters with 13.5 °C in average.

The National Water Agency of Brazil (ANA) provides hydro-meteorological data in the watershed since 2009. There are five pluviometers and two rules for river level measurements, providing data with the frequency of once and twice a day, respectively. The Agency also provides some samples of water quality data, such as turbidity, and of discharge, needed for discharge rating curve.

Predominant soils in the basin are the Cambisol, Red-Yellow Argisol, Neptosol and Fluvisol (Fig. 6). Red-Yellow Argisol consists predominantly of clay, while the Fluvisol and Neptosol of sand. Neptosol is shallower with rocky outcrops, while the Argisols and Cambisols are relatively deep, but also with stony and rockiness characteristics.

**HESSD**

doi:10.5194/hess-2015-490

**Effect of roads in InVEST sediment and streamflow simulation**

S. I. Saad et al.

## 2.2 InVEST Model

We used *Reservoir Hydropower Production and Avoided Reservoir Sedimentation Models of InVEST* (*Integrated Valuation of Environmental Services and Tradeoffs*, Sharp et al., 2014), to simulate stream and sediment flow in Posses river basin. InVEST consists of a suite of models, GIS-based, and uses climate and soil properties rasters as inputs. The models generally are composed by a biophysical component and an environmental evaluation component, which converts the former into environmental services and economic benefits. In this work we used only the biophysical components, which model water and sediment export, respectively, along the watershed. From now on, we will refer to these as Hydrological model and Sediment export model.

### 2.2.1 Hydrological model

We used Version 3.1 of InVEST Hydrological model (Sharp et al., 2014). It is a distributed hydrological model, and it is based on the annual water balance, in which, the water yield in each grid point is calculated by the difference between precipitation and actual evapotranspiration. Users have to provide information such as maps of land use and land cover, precipitation, potential evapotranspiration, soil depth and Plant Available Water Content (PAWC), besides crop factor ($K_c$) and root depth information. Thus the model calculates actual evapotranspiration by Zhang (2004) formulation, and, at last, water yield. Another parameter needed for Zhang formulation in InVEST is an empirical $Z$ parameter (Donohue et al., 2012).

For Posses simulations, we used land use and land cover map showed in Fig. 3, with parameter and input data summarized in Table 1. Pasture $K_c$ was achieved by calibration. $Z$ parameter was estimated by the number days of rain divided by 5 (Hamel and Guswa, 2015), which resulted in 25.

We simulated four hydrological years, matching to the period with observed data of river level and precipitation. In the region, the hydrological period goes from October to September of the next year, so that the period simulated went from 1 October 2010

Discussion Paper | Discussion Paper | Discussion Paper | Discussion Paper

**HESSD**

doi:10.5194/hess-2015-490

**Effect of roads in InVEST sediment and streamflow simulation**

S. I. Saad et al.

to the 30 September 2014. We will be referencing the hydrological years as hy2010-2011, hy2011-2012, hy2012-2013 and hy2013-2014, and each one represented one yearly simulation. Precipitation maps were computed using the five pluviometers data and Cressman Analysis (Fig. 7).

To estimate potential evapotranspiration maps, we used data from the Monte Verde meteorological station from Meteorology National Institute (INMET), located in the city of Camanducaia, MG, 30 km away from the basin and with an altitude of 1545 m. To consider the difference of altitude with Monte Verde station, and also the variation along the watershed itself, we applied a correction in Monte Verde temperature based on the temperature and difference of altitude, according to Lapse-Rate Model (Eq. 1), used for interpolation of station data and downscaling applications (Gao et al., 2012).

$$T_{cor}[°C] = T_{ref}[°C] + \Gamma[°C\,km^{-1}] \cdot \Delta h[km] \tag{1}$$

where $T_{cor}$ is the corrected temperature and $T_{ref}$ is the reference temperature, $\Gamma$ is the Standard Atmosphere Lapse-Rate, of $6.5\,°C\,km^{-1}$ and $\Delta h$ is the difference of altitude to the reference station.

Potential evapotranspiration was estimated by Penman–Monteith Method with SPEI package (Begueria and Serrano, 2013) from R-Cran Software, with the following data: Minimum and maximum temperature, solar radiation, solar radiation in the top of atmosphere, wind intensity in 2 m, altitude and latitude. Except for temperature, all the meteorological data used was obtained directely from Monte Verde Station. Figure 8 shows the computed evapotranspiration map, which showed similar pattern from mean temperature, with lower temperature and evapotranspiration rates in the higher parts of the basin. The mean potential evapotranspiration in the basin was 1123 mm.

### 2.2.2 Sediment export model

For sediment export model, we used version 2.4 of InVEST (Tallis et al., 2011). The model simulates soil loss in each grid point in a year, the amount of this soil that is

**HESSD**

doi:10.5194/hess-2015-490

**Effect of roads in InVEST sediment and streamflow simulation**

S. I. Saad et al.

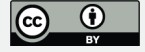

retained downstream, and how much is exported out of the watershed. Soil loss is calculated by Universal Soil Loss Equation (USLE) (Eq. 2), which includes information related to land use, soil properties and climate:

$$\text{USLE} = R \cdot K \cdot \text{LS} \cdot C \cdot P \tag{2}$$

where USLE is the annual soil loss ($\text{t ha}^{-1}\,\text{year}^{-1}$), $R$ is the rainfall erosivity factor, $K$ is the soil erodibility factor, LS is the slope length-gradient factor, $C$ is the cover management factor and $P$ is the support practice factor.

From the soil loss in each model grid point, a part is retained by the vegetation downstream and the other part is released. This retention capacity depends on the vegetation characteristic, and in InVEST, it is given by the Sediment Retention Efficiency parameter, required for each land-use kind. Lastly, InVEST account for the total exported sediment from the watershed in $\text{t year}^{-1}$. For our study, we divided it by the area of the watershed to characterize the sediment flow.

Table 3 sums up all the parameters and data used in the model. Digital Elevation Model was interpolated to 5 m from ASTER (Advanced Spaceborne Thermal Emission and Reflection Radiometer), of 30 m of resolution. Soil erodibility was obtained from Zolin et al. (2014), rain erosivity was calculated using monthly precipitation data from the gauges in the watershed, using Lombardi Neto and Moldenhauer (1980) relation (Eq. 3).

$$R = 67.355 \left( \frac{r^2}{p} \right)^{0.85} \tag{3}$$

where $R$ is the rain erosivity factor ($\text{MJ mm ha}^{-1}\,\text{year}^{-1}$), $r$ is monthly precipitation (mm), $P$ is annual precipitation (mm). The 12 monthly erosivity values are summed to the annual erosivity.

Threshold flow accumulation parameter, which defines the density of the streams, was defined by comparing stream outputs from InVEST with the observed net of rivers

Discussion Paper | Discussion Paper | Discussion Paper | Discussion Paper

**HESSD**

doi:10.5194/hess-2015-490

**Effect of roads in InVEST sediment and streamflow simulation**

S. I. Saad et al.

in the watershed (Fig. 9). Slope threshold, a parameter which limits LS calculations to two different models (for the lower slopes, the model of Renard et al. (1997) and for the greater, Yanhe and Chenglong, 1993), was set to zero, so that only the second model was used (Fig. 10). The reason for it is that the first one resulted in very high values of LS, and the mixture of the two equations led to an descontinuity in LS as a function of slope. Forest sediment retention efficiency was choseen comparing traping efficiency of a vegetative buffer from White and Arnold (2009), Liu et al. (2003), Park et al. (2011), and Yuan et al. (2009). So the value used was of 55 %. The pasture sediment retention efficiency was chosen according to calibration. More details on this parameters, as well as model sensitivity analysis will be provided in further publication.

### 2.2.3 Roads simulations

Posses watershed has 115 km of unpaved roads and present some characteristics favourable to sediment production in the basin, which make them essential in a sediment flow modelling:

1. High sediment production capacity

2. Inability to trap sediment

3. Alter water path after precipitation events, causing part of the water slide along the roads themselves

To solve item 1, we used the cover management and support practice factor (CP, Eq. 2) equals to 1, which represents the null contribution of roads to reduce soil loss in USLE. For the second item, we set the road sediment retention parameter to zero, representing no ability to retain sediment from the upstream areas.

The original DEM used for the simulations has no sufficient resolution for roads definition, so we used an artifice to solve the third item: the lowering of the DEM in the areas of the roads (a GIS process called as DEM burning). With roads slightly lower than the adjacent regions, part of the flow goes toward them, increasing the loss of

**HESSD**

doi:10.5194/hess-2015-490

**Effect of roads in InVEST sediment and streamflow simulation**

S. I. Saad et al.

Discussion Paper | Discussion Paper | Discussion Paper | Discussion Paper |

soil erosion. Therefore, we performed some experiments increasing the lowering of the roads (*barraginhas* DEM was also lowered in 10 m). We noticed that soil loss increased with the lowering of the roads (Fig. 11a), total retention decreased (Fig. 11b), what reflected in the increase of sediment flow (Fig. 11c).

Another effect of lowering the roads in the InVEST simulations is the increasing efficiency of *barraginhas*, strategically built near the steeper curves of the roads. With the increased flow on roads, the amount of sediments captured by *barraginhas* increases, and so their contribution by increasing total sediment retention. *Barraginhas* efficiency in sediment retention was used to determine the lowering in level of roads (Fig. 12). Thus the chosen lowered level was 1.0 m, equivalent to the level that retention reaches 63 % of the maximum, (analogous to e-folding time), and it is also consistent with the actual conditions of Posses' roads.

## 2.3 Calculation of sediment flow from observational data

To calculate streamflow, we used river level data in the mouth of Posses provided by ANA that are measured twice a day: at 7 a.m. and 5 p.m. (local time). We used the discharge rating curve of Mota da Silva et al. (2016) (Eq. 4).

$$Q = \begin{cases} 12.32\,(h[\text{m}] - 0.26)^{2.29}; & \text{for } h < 0.7\,\text{m} \\ 4.27\,h[\text{m}] - 1.09; & \text{for } h \geq 0.7\,\text{m} \end{cases} \tag{4}$$

where $Q$ is streamflow and $h$ is the level of the river.

Some measurements of turbidity and streamflow are also provided (Fig. 13). Based on this data and on Strauch et al. (2013), we described turbidity as function of streamflow, with power ratios and a linear relation (Fig. 14, Eq. 5).

$$\text{TU[NTU]} = \begin{cases} 94.582 \cdot Q^{1.1348}; & \text{for } Q < 0.23 \\ 308.92 \cdot Q^{1.9096}; & \text{for } 0.23 \leq Q < 0.95 \\ 560.42 \cdot Q - 252.3; & \text{for } Q \geq 0.95 \end{cases} \tag{5}$$

**HESSD**

doi:10.5194/hess-2015-490

**Effect of roads in InVEST sediment and streamflow simulation**

S. I. Saad et al.

Discussion Paper | Discussion Paper | Discussion Paper | Discussion Paper

where TU is turbidity and $Q$ is the streamflow ($m^3 s^{-1}$). For suspended sediment concentration, we used the relation of Lima et al. (2011):

$$SS[mg L^{-1}] = 1.114\ TU[NTU] + 1.4731 \tag{6}$$

where SS is suspended sediment concentration.

Then, we calculated sediment export (SE, $t\,year^{-1}$) by:

$$SE[t\,year^{-1}] = 31.536 \cdot Q[m^3 s^{-1}] \cdot SS[mg L^{-1}] \tag{7}$$

Then, we calculated sediment export (SE, $t\,year^{-1}$) by:

Sediment flow is given by the ratio between sediment export and drainage area:

$$SF[t\,km^{-2}\,year^{-1}] = (31.536 \cdot Q[m^3 s^{-1}] \cdot SS[mg L^{-1}])/A_u[km^2] \tag{8}$$

where $A_u$ is the upstream area and SF is sediment flow.

Figure 15 illustrates the flowchart of sediment flow calculation, starting from observed streamflow.

## 2.4 Calculation of sediment concentration and turbidity from simulations

At first glance, one would obtain sediment concentration as function of simulated sediment flow and streamflow just by inverting Eq. (8) so that sediment concentration will be given by Eq. (9), according to the flowchart in Fig. 16.

$$SS[mg L^{-1}] = \frac{SF[t\,km^{-2}\,year^{-1}] \cdot A_u[km^2]}{31.536 \cdot Q[m^3 s^{-1}]} \tag{9}$$

However, apart from observation data, simulation data are annually based, and so the bi-linear relation of sediment concentration with sediment and streamflow may result in some differences. These differences are illustrated in Fig. 17, which shows: (1) sediment concentrations based on twice a day observational data, (2) the average of it,

Discussion Paper | Discussion Paper | Discussion Paper | Discussion Paper

**HESSD**

doi:10.5194/hess-2015-490

**Effect of roads in InVEST sediment and streamflow simulation**

S. I. Saad et al.

and (3) "average" sediment concentration calculated directly from average sediment flow and average streamflow. Our hypothesis is that concentration estimated from simulations by Eq. (9) is related to the computation (3), while the "right" one is related to the computation (2). Figure 18 shows concentration estimated from observation using computation (2) (concentration integrated daily in a year) and computation (3) (given by annual sediment and streamflow). One can notice that computation (3) is much higher than (2), reaching values almost 20 times higher. While computation (2) represents the "truth" and (3) represents the mistaken that may be committed if using Eq. (9) in an annual analysis, we added a constant in the equation, calculated by the average of the ratio between second and third computation in each year in Fig. 18. We called this constant temporal granularity coefficient as it accounts by the difference between calculating concentration with daily and annual data. Equation (9) was rewritten as Eq. (10):

$$SS[mg\,L^{-1}] = ctg\frac{SF[t\,km^{-2}\,year^{-1}] \cdot A_u[km^2]}{31.536 \cdot Q[m^3\,s^{-1}]} \tag{10}$$

where ctg = 0.12 is temporal granularity coefficient.

As for turbidity, no further correction had to be provided due to its linear relation with concentration (Eq. 5).

## 2.5 Calibration method

Calibration was performed in terms of streamflow and sediment flow, using estimated data from observation. As an indicator of the error of simulations we used the relation between volumes (Eq. 11). Each one of the four hydrological years represented one time step.

$$\Delta V = \frac{\sum_{t=1}^{N} p_t - \sum_{t=1}^{N} o_t}{\sum_{t=1}^{N} o_t} \tag{11}$$

Discussion Paper | Discussion Paper | Discussion Paper | Discussion Paper |

**HESSD**

doi:10.5194/hess-2015-490

**Effect of roads in InVEST sediment and streamflow simulation**

S. I. Saad et al.

where $\Delta V$ is the relation between volumes, $N$ is the number of years, $p_t$ is simulated value in time step $t$ and $o_t$ is the observed value.

Basically, streamflow calibration consisted in altering pasture $K_c$ gradually so that $\Delta V$ in Eq. (11) was set to its minimum. For the sediment flow calibration, the chosen
5 parameter was pasture sediment retention efficiency.

## 3 Results

### 3.1 Estimated flows from observational data

For each data of river level, measured twice a day, we estimated streamflow, turbidity, sediment concentration, and sediment flow. Figure 19 shows the monthly mean tem-
10 poral series for each variable and its monthly dispersion, represented by the standard deviation of the measures. A seasonal pattern in the streamflow can be noticed, with higher values usually in January, at the peak of the rainy season. Standard deviation often exceeded the seasonal range of streamflow, what is due to the small extent of the basin, whose streamflow is quickly affected by local rainfall. Turbidity shows similar pat-
15 tern, except for the lower values of streamflow, which generate a relative low turbidity. Sediment concentration showed a linear pattern with turbidity, and ultimately, sediment flow was predominantly influenced by extremes, due to its bi-dimensional dependency of sediment concentration and streamflow (Eq. 7).

Regarding the annual average of these flows (Fig. 20), it is also clear the proportion-
20 ality relationship between the variables. In exception, the streamflow of the hydrological year 2010–2011 is slightly lower than that of 2010–2011, while the turbidity of the first hydrological year is slightly higher than the second. This occurred because, although the average streamflow was higher in the second hydrological year, the first year had higher peak flows (Fig. 19), which increased turbidity.
25 These measures are strongly dependent on heavy rain events, featuring peaks in the variables, and influencing its annual averages. It is worth noting the great differ-

Discussion Paper | Discussion Paper | Discussion Paper | Discussion Paper |

**HESSD**

doi:10.5194/hess-2015-490

**Effect of roads in InVEST sediment and streamflow simulation**

S. I. Saad et al.

ence between the mean and the median, respectively, since mean is sensitive to extremes, particularly to precipitation events: Streamflow: 193 and 141 L s$^{-1}$; turbidity: 32 and 10 NTU; sediment concentration: 37 and 12 mg L$^{-1}$; and sediment flow: 134 and 5 t km$^{-2}$ year$^{-1}$.

The median turbidity (10 NTU) is above the minimum standard for drinking water, which is 5 NTU (SABESP, 2014). Above 50 NTU, the reproduction of numerous species of fish is quite impaired. However, both median and average were below this threshold.

Among the variables considered, sediment flow showed the highest differences between mean and median, representing a greater sensitivity to high flow rates. The mean value (134 t km$^{-2}$ year$^{-1}$) is within the expected range, whereas average estimates of Brazilian rivers are between 3 and 170 t km$^{-2}$ year$^{-1}$ (Lima et al., 2008). In Pipiripau (188 km$^2$) and Descoberto Lake (105 km$^2$) Rivers Basins, in Central Brazil, estimate ranges from 10 to 26 t km$^{-2}$ year$^{-1}$ (Strauch et al., 2013). For Europe, Vanmaercke et al. (2011) evaluated a number of sediment flow estimation studies across the continent and highlighted that, despite the high variation between different studies, there were marked differences between the different European climates and topographical features. In the flatter boreal climate zone, most of the flow sediments measurements were below 50 t km$^{-2}$ year$^{-1}$, while in Mediterranean and mountainous areas, most of the measures exceeded 200 t km$^{-2}$ year$^{-1}$.

## 3.2 Simulated flows

Figure 21 shows the simulated and observed flows. Due to calibration, the relation between volumes (Eq. 3) was low (5 and 1 %, respectively for streamflow and sediment flow). The years with higher observed streamflow correspond to the years with higher simulated streamflow, and the same was true for the sediment flow. The sediment flow of simulations were softer than of the observations, as they underestimated the observations in the years with greater sediment flow (hy2010-2011 and hy2011-2012) and overestimated in the years of lower sediment flow (and hy2012-2013 hy2013-2014). However, in the last two hydrological years the Southeast of Brazil were marked by an

**HESSD**

doi:10.5194/hess-2015-490

**Effect of roads in InVEST sediment and streamflow simulation**

S. I. Saad et al.

historical drought (Coelho et al., 2015), and extreme events are often the weakness of the models.

The concentration pattern calculated from the simulations was notably different from the calculated from observations. In the 2011–2012 hydrological year, for example, the concentration was the lowest among the other years, even representing the second year with the greatest sediment flow. As for 2013–2014 hydrological year, it showed the highest concentration, even though being the year with the lowest sediment flow. This occurred because the decrease in streamflow was higher than the decline of sediment flow compared to previous years. The turbidity showed a similar behaviour due to its linear relationship with concentration. These low correlations in annual concentration, may be consequence of the high uncertainty associated with concentration (Strauch et al., 2013), but it compromises an annual comparison, although succeeding in an annual mean analysis, with an error of 11 % (relation between volumes, Eq. 11).

## 3.3 Simulated spatial patterns

InVEST provides maps of soil loss and sediment exported, which represent the amount of soil produced in each grid cell that will reach the stream. Soil loss is remarkably higher in the roads (Fig. 22a), and it shows smaller values in grid cells covered by forest. Higher LS values (Fig. 10) also respond to higher soil loss. With respect to sediment exported (Fig. 22b), the higher values are related to the greater production in the roads, and some areas near the stream (Fig. 9) represent a second role, due to the smaller path travelled by the sediments until they reach the stream.

A second simulation was performed similar from the former, but with no roads. It was also calibrated in terms of sediment flow (Fig. 23). In this case the spatial pattern in soil loss is very different from the former. In this case, the differences between areas of pasture and forest are much stronger. Moreover, the average in this case is higher due to the calibration process, made in terms of sediment flow in the mouth of the watershed.

Discussion Paper | Discussion Paper | Discussion Paper | Discussion Paper |

**HESSD**

doi:10.5194/hess-2015-490

**Effect of roads in InVEST sediment and streamflow simulation**

S. I. Saad et al.

## HESSD

doi:10.5194/hess-2015-490

**Effect of roads in InVEST sediment and streamflow simulation**

S. I. Saad et al.

# 4   Simulated effects of roads and *barraginhas*

In order to evaluate the effect of roads and *barraginhas* in sediment export, we performed simulations with InVEST in four scenarios: (1) with roads, (2) with roads and *barraginhas*, (3) with no roads and no *barraginhas* (uncalibrated), and (4) calibrated with no roads and with *barraginhas*.

Figure 24 shows the sediment balance represented by total soil loss, total retention, and the sediment flow in the mouth of the watershed, and also, the retention within *barraginhas* and soil loss within roads domain. The subtraction between total soil loss and sediment retention should be close to sediment flow in the mouth of the watershed. By doing this we got errors not higher than 5 %, meaning that InVEST output variables are coherent in the issue of sediment balance. Comparisons between scenarios show a large control of roads in soil production. As compared to "No Roads" scenario, the "Roads" scenario showed a large increase of total soil loss, and a lower increase in total retention, and these caused the increase of sediment flow of more than 5 times with the roads, which represent about 2 % of total land use. In comparison, Fu et al. (2007) showed that approximately half of the sediment load was generated by the 2 % of land cover of roads in Moruya and Tuross-Deua basins in the southeast of Australia.

The scenario calibrated with no roads and with *barraginhas* presented higher total soil loss and retention, due to the calibration processes, as simulations with no roads have to show a higher soil loss to compensate its higher retention. The flow in the mouth was similar to the first two scenarios, but not identical whereas calibration processes were performed using none of this scenarios, but the current land use conditions of the catchment. Despite the increasing of total retention, retention in *barraginhas* decreased, showing its higher efficiency nearby the roads.

As to *Barraginhas* scenario, there was an increase of sediment retention in its domain, that resulted in a slightly increase of total retention and diminished in 2 % the sediment flow in the mouth. In comparison, Strauch et al. (2013) simulated 5.5 % of sediment reduction in the Pipiripau River watershed in central Brazil, with SWAT model.

They considered a density of 2.5 *barraginhas* per road kilometre, each one with an area of 50 m$^2$. In our case, Posses has a density of 0.76 *barraginhas* per road kilometre, and for InVEST we considered an area of 25 m$^2$, equivalent to the grid point size, and similar to observed conditions. The efficiency of *barraginhas* in our case was higher maybe due to its increased potential due to the lowering of roads, as seen in Fig. 12.

## 5  Conclusions

This paper showed the implementations of hydrology and sediment export models of InVEST in Posses, a small river basin in Brazil, but important due to its contribution to Cantareira water supply system and principally because it represents the first Brazilian experience of Payment of Environmental Services (PES), supported by a municipal low. Simulations were performed with high resolution (5 m × 5 m), so that some particularities of the watershed such as roads and *barraginhas*, small sediment retention basins nearby the roads, could be taken into account. The roads played an important role in sediment export, as it was increased in 5 times in a simulation with roads as compared to another with no roads. *Barraginhas* increased the total sediment retention, and decreased sediment export by 2 %.

One challenge faced for implementing InVEST was to find data for sediment flow calibration. The watershed is provisioned by twice-a-day measures of the river level, and few samples of the water turbidity and streamflow. Based on the samples, we adjusted a rating curve of turbidity as function of streamflow, and then we used other references to calculated sediment concentration, and lastly sediment flow. This estimative is very sensitive to high streamflow events, and so to streamflow errors. Due to the small extension of the watershed, the level of the river, measured only twice a day, may miss some precipitation events, what increases uncertainties, not only in streamflow but principally on sediment flow. We believe these uncertainties will not disqualify the average sediment and stream flow estimations, as our further comparisons

**HESSD**

doi:10.5194/hess-2015-490

**Effect of roads in InVEST sediment and streamflow simulation**

S. I. Saad et al.

Discussion Paper | Discussion Paper | Discussion Paper | Discussion Paper

will be in mean terms. However, we raise the importance of further observation studies of turbidity, soil concentration and streamflow using high temporal resolution.

InVEST simulations provided sediment and streamflow, and we calculated sediment concentration and turbidity that will be used for further studies of water quality. For this purpose, we raised an important issue that imped the use of the former calculations, which is the difference between the frequency of observational data (twice a day) and simulation data (annual). Using these frequencies, the concentration estimated using annual data of sediment and streamflow would give 8 times greater than the one with daily data, and this correction was implemented in simulated sediment concentration. Results showed that concentration estimated by simulations presented an error of 11 % as comparing to concentrations estimated by observations, what can be considered a reasonable result considering the errors in the estimates by the own observations, but cannot be applied in a year-to-year comparison, as the correlations in the years have shown to be very low. On the other hand, calibration was performed in terms of streamflow and sediment flow succeeded, with errors of 5 and 1 %, respectively, and good correlation between simulations and observations.

The model built to estimate sediment flow data directly from streamflow, at first sight makes sediment flow simulations with InVEST unnecessary. However, one must keep in mind that it is an appropriate model for the mean and current state of the watershed. The model does not distinguish occasional increases in streamflow uncorrelated by increases in turbidity, which can occur for a particular characteristic or location of a precipitation event, or by the fact there is some change in land use and cover in the watershed. Thus, our motivation in using a model for simulating sediment flow is that it will make possible further studies of impact on land use change on sediment and streamflow and on associated environmental services such as the improvement on water quality or streamflow modification.

The absence of roads considerations in simulations would not affect final calibrated sediment flow in the mouth of the watershed due to compensation in calibration parameters. However, as the parameters used for calibration are specific for pasture, the

**HESSD**

doi:10.5194/hess-2015-490

**Effect of roads in InVEST sediment and streamflow simulation**

S. I. Saad et al.

main land use in the catchment, this would lead to a misrepresentation of the difference between pasture and forest. While roads have a great importance on sediment yield, the omission of the roads would force pastureland contributes to a higher level of sediment yield to match observed data in the calibration process. In the catchment, pastureland is the area for the cattle, with no management practices being held, and forest are the areas for environmental services delivery, and a good representation between them is crucial for further land use change studies. Moreover, with no representation of the roads, the simulations of the Best Management Practices such as the building of terraces and *barraginhas*, would probably be underrated.

*Acknowledgements.* The authors thank The Nature Conservancy (TNC) for providing data from the catchment. The first author thanks The National Council for Scientific and Technological Development (CNPq, Process 15936363/2012-8) for a doctoral fellowship, and the development InVEST team for technical support.

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

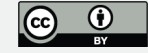

**Table 1.** Parameters and input data required for InVEST hydrological model.

| Information | Type | Value | Source |
|---|---|---|---|
| Land use | Map (raster) | Fig. 3 | Mixed |
| Precipitation | Map (raster) | Fig. 7 | Interpolated |
| ET0 | Map (raster) | Fig. 8 | Estimated from a nearby station |
| Soil Depth | Map (raster) | Table 2 and Fig. 6 | Azevedo (2008) |
| PAWC | Map (raster) | Table 2 and Fig. 6 | Azevedo (2008) |
| $K_c$ | per LULC | pasture: 0.8 forest: 1 eucaliptus: 1 | Calibration and Allen et al. (1998) |
| root depth | per LULC | pasture: 1500 mm forest: 4000 mm eucaliptus: 5000 mm | Canadell et al. (1996), Saad et al. (2010) |
| $Z$ | Constant | 25 | Estimated |

Discussion Paper | Discussion Paper | Discussion Paper | Discussion Paper | Discussion Paper |

**HESSD**

doi:10.5194/hess-2015-490

**Effect of roads in InVEST sediment and streamflow simulation**

S. I. Saad et al.

**Table 2.** Plant Available Water Content (PAWC), soil depth and erodibility ($K$ factor), by soil type. The first two are input data for InVEST hydrological model and the last for sediment export.

| Soil | PAWC (mm) | Soil Depth (mm) | Erodibility |
|------|-----------|-----------------|-------------|
| Red-Yellow Argisol | 0.048 | 3000 | 0.04 |
| Haplic Cambisol | 0.03 | 3000 | 0.035 |
| Humic Cambisol | 0.035 | 3000 | 0.0254 |
| Fluvisol | 0.05 | 4000 | 0.042 |
| Neptosol | 0.03 | 1000 | 0.046 |

**HESSD**

doi:10.5194/hess-2015-490

**Effect of roads in InVEST sediment and streamflow simulation**

S. I. Saad et al.

**Table 3.** Parameters and input data required for InVEST sediment export model. The second options for pasture parameter were set after an alternative calibration which did not consider the roads.

| Information | Type | Value | Source |
|---|---|---|---|
| Land use | Map (shape) | Fig. 3 | Mixed |
| DEM | Map (raster) | Fig. 4 | ASTER |
| $R$ Factor | Map (raster) | Fig. 7 | Calculated (Eq. 3) |
| $K$ Factor | Raster | Fig. 6 and Table 2 | Zolin et al. (2014) |
| $C$ Factor | per LULC class | pasture: 0.055 | Silva et al. (2010) and calibration |
| | | forest: 0.001 | Silva et al. (2010) |
| | | eucaliptus: 0.0013 | Martins (2005) |
| Sediment Retention Efficiency | per LULC class | pasture: 36 % | Calibration |
| | | forest: 55 % | Estimated by other references |
| | | eucaliptus: 40 % | Estimated |
| Threshold Flow accumulation | Constant | 2000 | Comparison of net of rivers |
| Slope Threshold | Constant | 0 | Adjusting LS factor |

**HESSD**

doi:10.5194/hess-2015-490

**Effect of roads in InVEST sediment and streamflow simulation**

S. I. Saad et al.

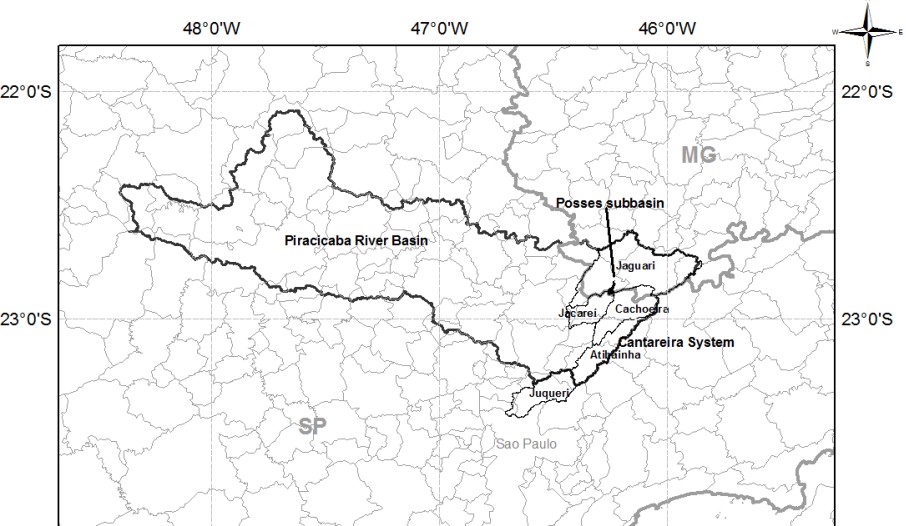

**Figure 1.** Location of Posses watershed, subbasin of Jaguari and of Piracicaba basin. Cantareira water supply system and its basins were also highlighted.



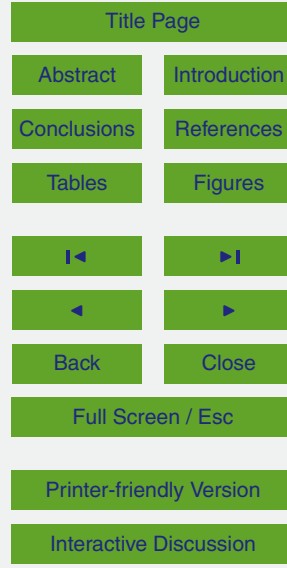

## HESSD

doi:10.5194/hess-2015-490

**Effect of roads in InVEST sediment and streamflow simulation**

S. I. Saad et al.



**Figure 2.** Location of *barraginhas* and roads. The shapes were expanded for clarity.

**HESSD**

doi:10.5194/hess-2015-490

**Effect of roads in InVEST sediment and streamflow simulation**

S. I. Saad et al.

Forest (21.9%)
Eucaliptus (1.8%)
Pasture (71.2%)
Water (0.1%)
Barraginhas (0.1%)
Roads (2%)
Growing Forest (2.9%)

0  0.3 0.6    1.2 km

**Figure 3.** Land use and land cover map of Posses.

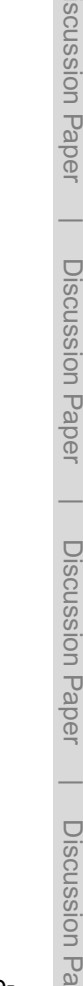

**Figure 4.** Digital Elevation Model ASTER (Advanced Spaceborne Thermal Emission and Reflection Radiometer), with the 30 m of resolution, and net of rivers.

Discussion Paper | Discussion Paper | Discussion Paper | Discussion Paper

**HESSD**

doi:10.5194/hess-2015-490

**Effect of roads in InVEST sediment and streamflow simulation**

S. I. Saad et al.

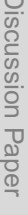

**HESSD**

doi:10.5194/hess-2015-490

**Effect of roads in InVEST sediment and streamflow simulation**

S. I. Saad et al.

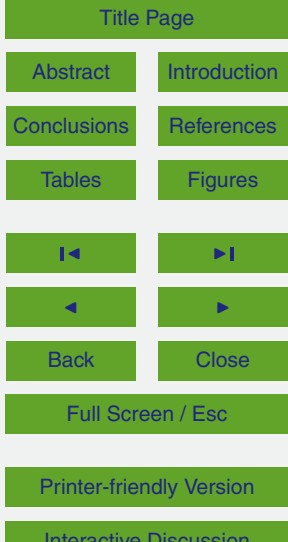

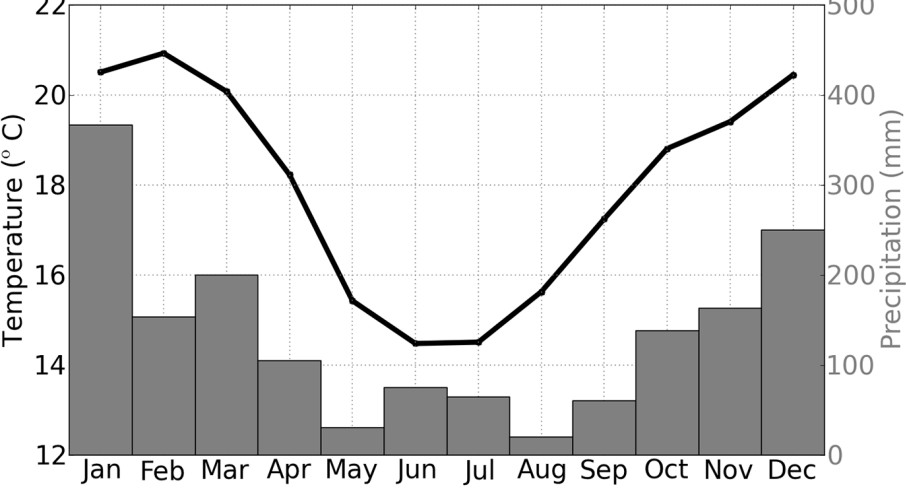

**Figure 5.** Mean monthly temperature and precipitation in Posses. Temperature was estimated from Monte Verde meteorological station, and the precipitation from pluviometers in the watershed.

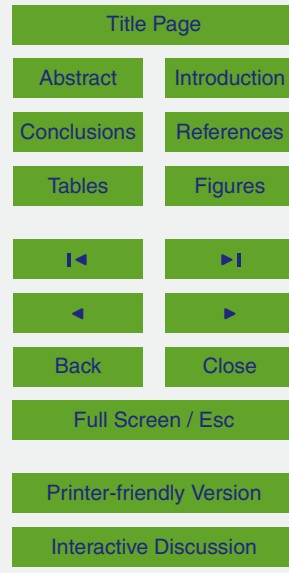

## HESSD

doi:10.5194/hess-2015-490

**Effect of roads in InVEST sediment and streamflow simulation**

S. I. Saad et al.

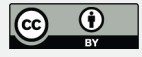

**Figure 6.** Soil map in Posses. Adapted from Freitas et al. (2008).



**Figure 7.** Annual mean precipitation interpolated by Cressman method (shaded) from the rain gauges (black points) from October 2010 to September 2014.

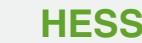

**HESSD**

doi:10.5194/hess-2015-490

**Effect of roads in InVEST sediment and streamflow simulation**

S. I. Saad et al.

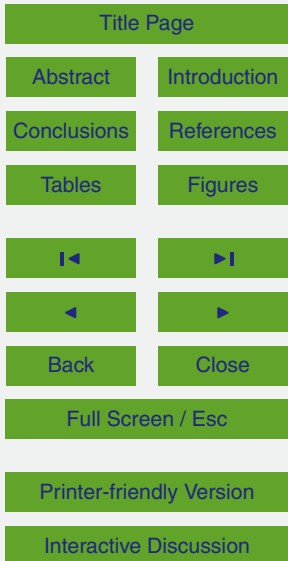

**HESSD**

doi:10.5194/hess-2015-490

**Effect of roads in InVEST sediment and streamflow simulation**

S. I. Saad et al.

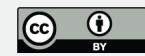

**Figure 8.** Annual mean reference evapotranspiration in Posses obtained by the Penman–Monteith method.

Discussion Paper | Discussion Paper | Discussion Paper | Discussion Paper


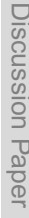

**Figure 9.** Net of rivers in Posses generated with InVEST and generated from field research. The former was provided by TNC (The Nature Conservancy).

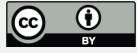

HESSD

doi:10.5194/hess-2015-490

Effect of roads in InVEST sediment and streamflow simulation

S. I. Saad et al.

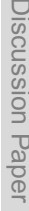

**Figure 10.** Map of the slope length-gradient factor (LS), generated with InVEST.

Discussion Paper | Discussion Paper | Discussion Paper | Discussion Paper

**HESSD**

doi:10.5194/hess-2015-490

**Effect of roads in InVEST sediment and streamflow simulation**

S. I. Saad et al.

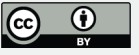

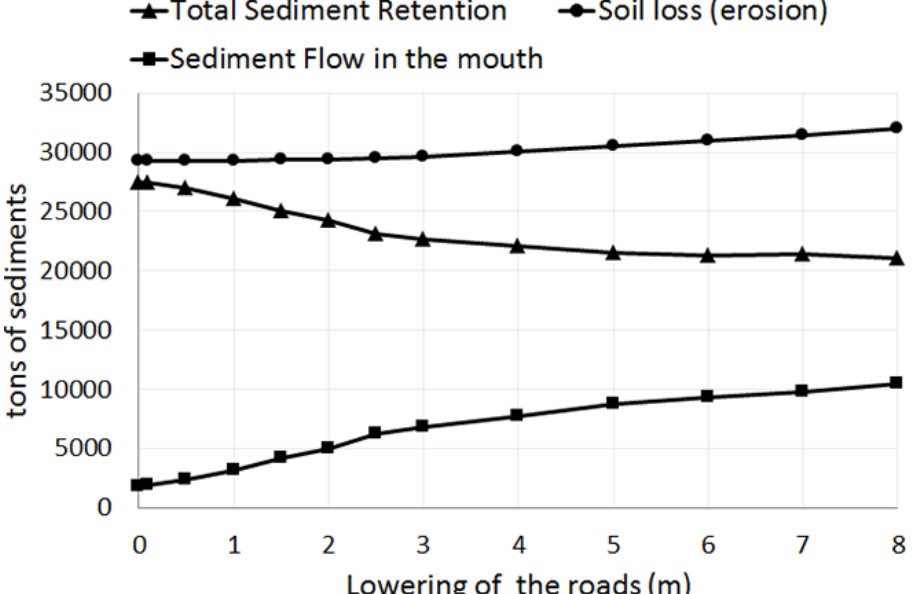

**Figure 11.** Total sediment retention, soil loss and sediment flow in the mouth of the watershed (t), as functions of lowering of DEM over the roads.

**HESSD**

doi:10.5194/hess-2015-490

**Effect of roads in InVEST sediment and streamflow simulation**

S. I. Saad et al.

Discussion Paper | Discussion Paper | Discussion Paper | Discussion Paper

**HESSD**

doi:10.5194/hess-2015-490

**Effect of roads in InVEST sediment and streamflow simulation**

S. I. Saad et al.

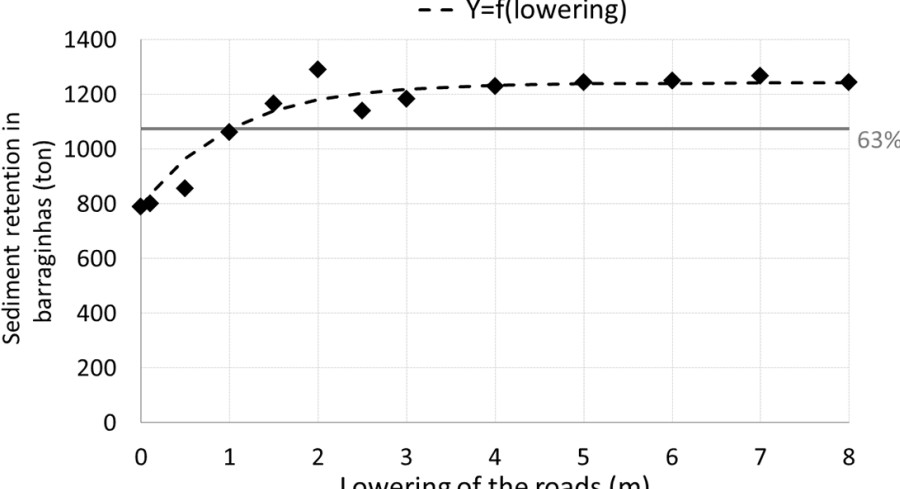

**Figure 12.** Sediment retention in barraginhas as function of the lowering of the roads. The line in gray indicates retention of 63 % of the maximum.

Discussion Paper | Discussion Paper | Discussion Paper | Discussion Paper

**HESSD**

doi:10.5194/hess-2015-490

**Effect of roads in InVEST sediment and streamflow simulation**

S. I. Saad et al.

**Figure 13.** Some samples of **(a)** streamflow and **(b)** turbidity, in Portal das Estrelas and in the Mouth of Posses.



Discussion Paper | Discussion Paper | Discussion Paper | Discussion Paper | Discussion Paper

**HESSD**

doi:10.5194/hess-2015-490

**Effect of roads in InVEST sediment and streamflow simulation**

S. I. Saad et al.

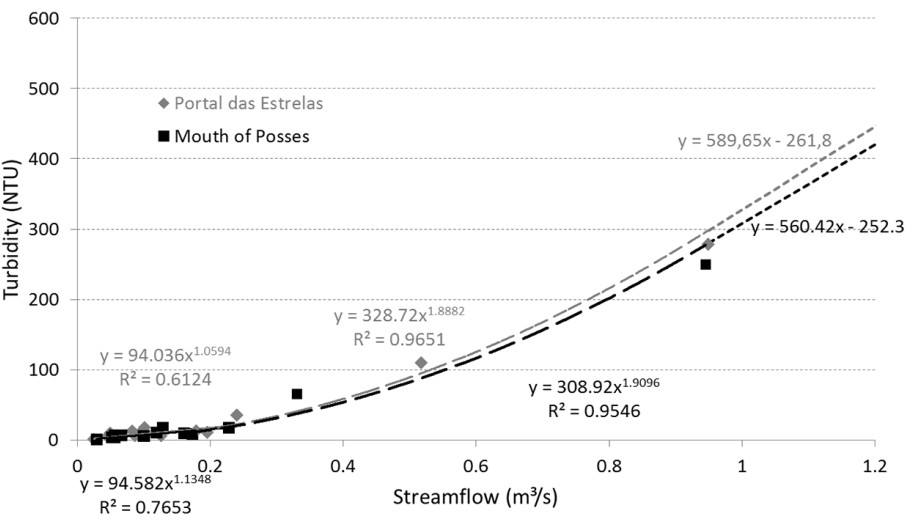

**Figure 14.** Turbidity as function of streamflow for the same samples of Fig. 13, in Portal das Estrelas and Mouth of Posses, and their power curves and linear relation fitted to the data.

# From the observations

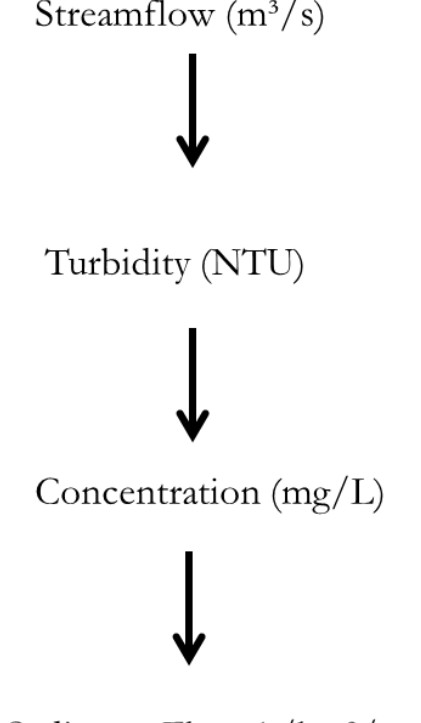

Streamflow (m³/s)

Turbidity (NTU)

Concentration (mg/L)

Sediment Flow (t/km²/year)

**Figure 15.** Flowchart of calculations starting from the observed streamflow. The formulations used for each variable were: streamflow: Eq. (4); turbidity: Eq. (5); concentration: Eq. (6); sediment flow: Eq. (8).

Discussion Paper | Discussion Paper | Discussion Paper | Discussion Paper

**HESSD**

doi:10.5194/hess-2015-490

**Effect of roads in InVEST sediment and streamflow simulation**

S. I. Saad et al.



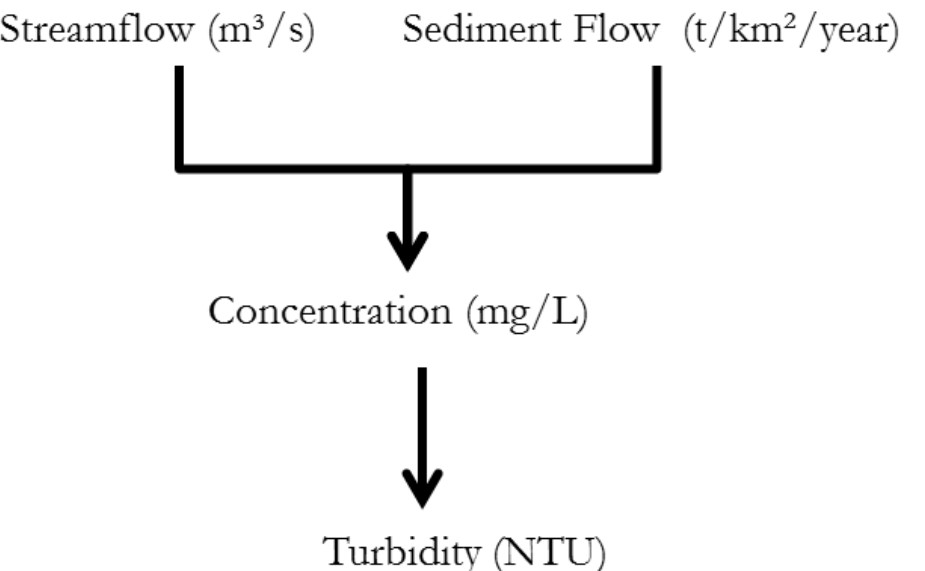

**Figure 16.** Flowchart of calculations stariting from simulated streamflow and sediment flow. Formulations used for each variable were: concentration Eq. (10); turbidity: Eq. (6).

Discussion Paper | Discussion Paper | Discussion Paper | Discussion Paper | Discussion Paper |

**HESSD**

doi:10.5194/hess-2015-490

**Effect of roads in InVEST sediment and streamflow simulation**

S. I. Saad et al.

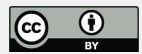

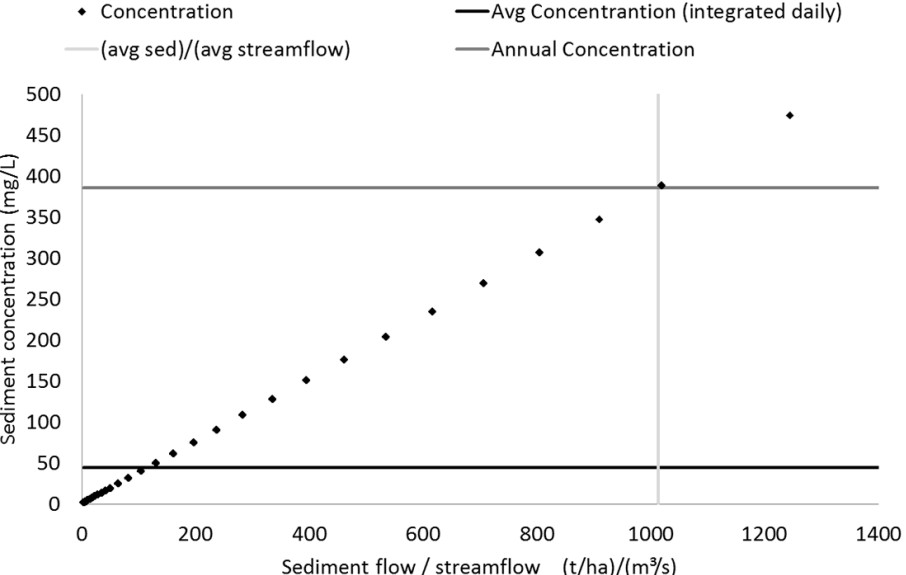

**Figure 17.** Calculated sediment concentration as function of the ratio (sediment flow)/streamflow (points), average of concentration (black line), ratio of (average of sediment flow)/(average streamflow) (vertical line), and concentration computed with the former (annual concentration, gray horizontal line).

Discussion Paper | Discussion Paper | Discussion Paper | Discussion Paper |

**HESSD**

doi:10.5194/hess-2015-490

**Effect of roads in InVEST sediment and streamflow simulation**

S. I. Saad et al.

Discussion Paper | Discussion Paper | Discussion Paper | Discussion Paper |

**HESSD**

doi:10.5194/hess-2015-490

**Effect of roads in InVEST sediment and streamflow simulation**

S. I. Saad et al.

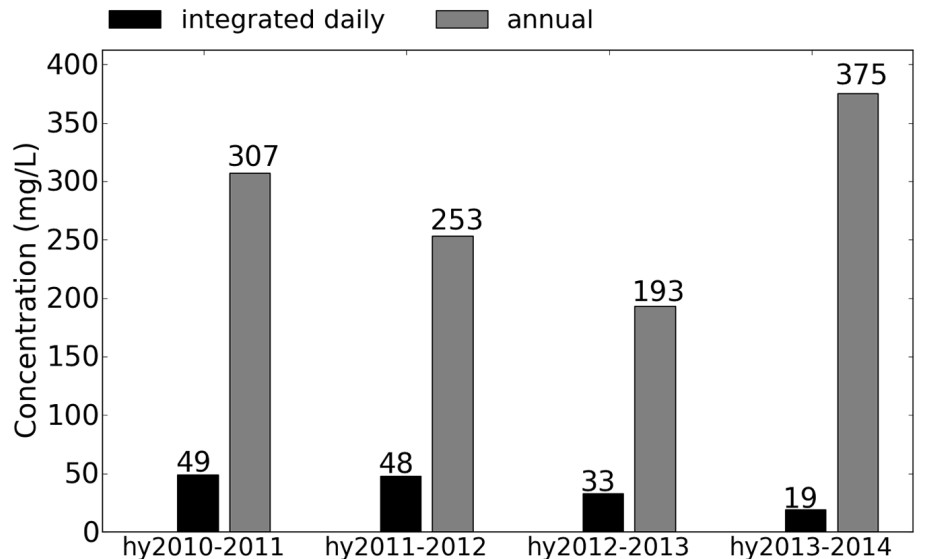

**Figure 18.** Average of sediment concentration (intagrated daily, in black), and annual concentration as function of annual sediment flow and annual streamflow (in gray).

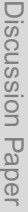

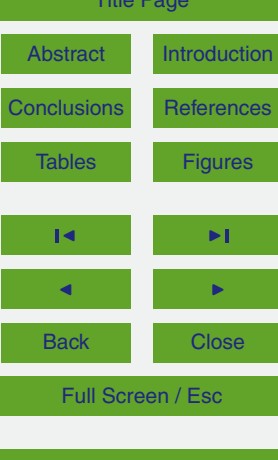

**HESSD**

doi:10.5194/hess-2015-490

**Effect of roads in InVEST sediment and streamflow simulation**

S. I. Saad et al.

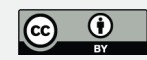

**Figure 19.** Temporal series of streamflow, turbidity, sediment concentration and sediment flow at the mouth of Posses, from October 2010 to September 2014. The black line represents the monthly average and the shaded in gray the standard deviation between the measurements of each month.

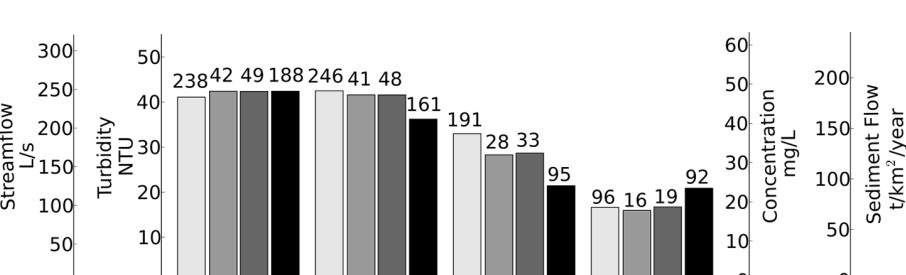

**Figure 20.** Average per hydrological year of streamflow, turbidity, sediment concentration and sediment flow (at this order) at the mouth of Posses. The last three were calculated as function of the twice-a-day streamflow.

Discussion Paper | Discussion Paper | Discussion Paper | Discussion Paper

**HESSD**

doi:10.5194/hess-2015-490

**Effect of roads in InVEST sediment and streamflow simulation**

S. I. Saad et al.

**HESSD**

doi:10.5194/hess-2015-490

**Effect of roads in InVEST sediment and streamflow simulation**

S. I. Saad et al.

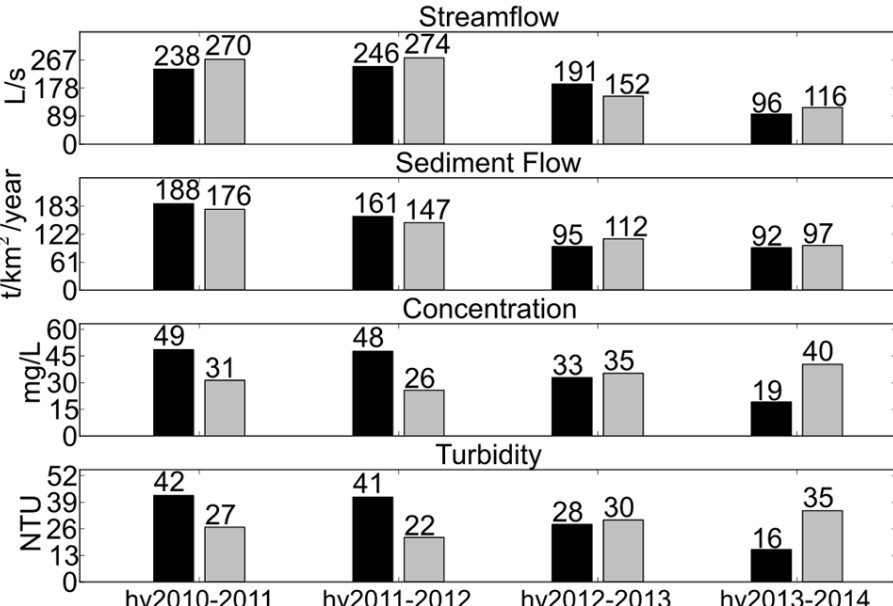

**Figure 21.** Average per hydrological year of streamflow, sediment flow, concentration, and turbidity estimated from observations and from simulations in Posses.

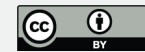

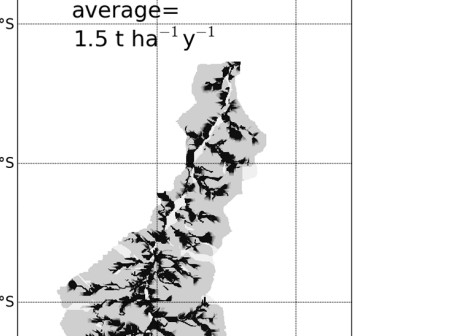

**Figure 22. (a)** Soil loss and **(b)** sediment export simulated with InVEST in the 2011–2012 hydrological year.

**HESSD**

doi:10.5194/hess-2015-490

**Effect of roads in InVEST sediment and streamflow simulation**

S. I. Saad et al.

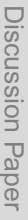

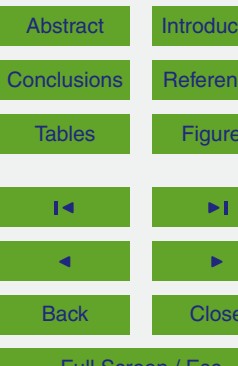

Soil loss (t ha$^{-1}$ y$^{-1}$) hy2011-2012

average=
27 t ha$^{-1}$ y$^{-1}$

**Figure 23.** Soil loss simulated with InVEST, not considering the roads, in the 2011–2012 hydrological year. Sediment export was not shown due to its similarities with former simulations.

**HESSD**

doi:10.5194/hess-2015-490

**Effect of roads in InVEST sediment and streamflow simulation**

S. I. Saad et al.

Discussion Paper | Discussion Paper | Discussion Paper | Discussion Paper

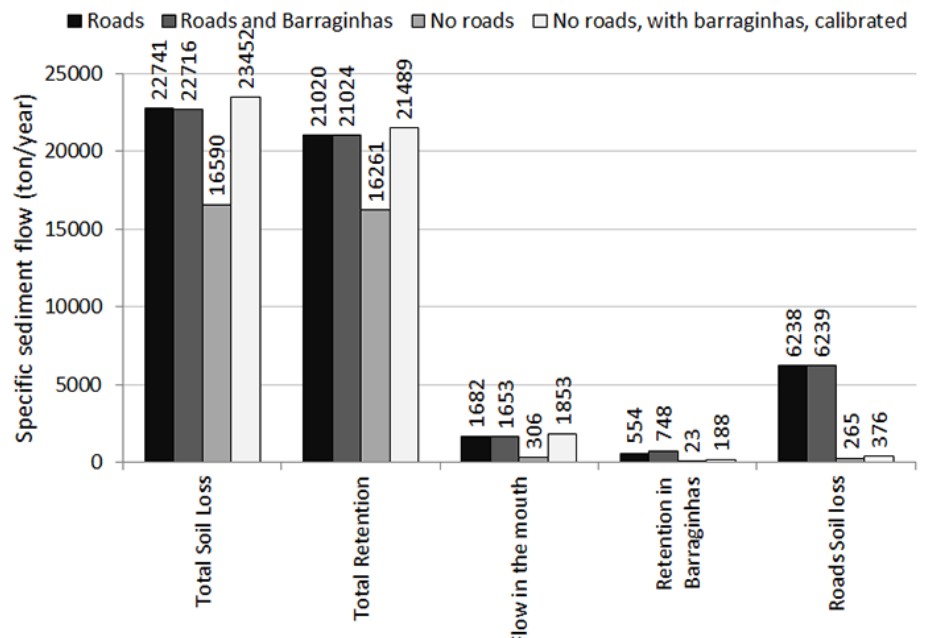

**Figure 24.** Specific flows of sediment: total soil loss, total sediment retention, flow in the mouth of the watershed, retention in *barraginhas*, and Roads soil loss, for four land use and land cover scenarios: with the roads, with roads and *barraginhas*, with no roads and no *barraginhas*, calibrated with no roads and with *barraginhas*.

Discussion Paper | Discussion Paper | Discussion Paper | Discussion Paper

**HESSD**

doi:10.5194/hess-2015-490

**Effect of roads in InVEST sediment and streamflow simulation**

S. I. Saad et al.

