# Peer review of "The impact of roads and sediment basins on simulated river discharge and sediment flux in an experimental catchment designed to improve ecosystem services"

_Hydrology and Earth System Sciences, 2015_

## Referee Comment (RC1) · Anonymous Referee #1 · 21 Jan 2016

Review comments on HYDRO-D-15-00236

Hydrologic and erosion models require spatial information on climate, topography, soils and land-use but also on landscape key features such as terraces, ditches, roads, dams and other structures that play an important role in water movements. The impact of these structures on water and sediment outputs from river basins together with the sensitivity analysis of models to the density of these spatial input data has thus become a challenging issue in environmental research and modeling. This is where hessd-12-4387-2015 comes in where the authors evaluated effects of dams and roads on hydrological simulation by the InVEST model.

**While I found the subject very interesting with aim towards payment for ecosystem services, the efforts the authors have put in data analysis, writing and the illustrations, this paper does not meet the requirements of international journals.**

There is a lack of acknowledgment of the existing literature on the field of research, which is usually followed by the indentification of research gaps and research questions. None of this has been performed as can be seen in the abstract and Introduction sections.

This document appears most likely to be one of the thousands documents brasilian students send to international journal, as a compulsory step for granting their prepared degree. In the present case however, the English terms and grammar seem to have been edited by an native English speaker or a professional.

**My suggestion to the authors is to consider the following tips for scientific writing,** mainly on the structure on the different sections of a research article. Moreover, **by reading the literature on the subject** (hundreds of international papers have been published on impact of roads, dams, … on water and sediment movements within river basins) the **authors will probably realize more is to be done in terms of the methods used (calibration and validation are compulsory in hydrological modeling) and presentation and discussion of data.**

Comments on abstract:

Line 1: the research proposed here is not per see on PES but on hydrological modeling. Research gaps should be presented here on the core of the paper
Line 3: what is "evidence-base management"?; the term "effect" contains strange "f" and this throughout the text
Line 6: what "comparison" is it about?
Line 9: the objectives of the study are missing
Lines 6-8: this statement is not really true. Models use landscape information as basic input.
Line 10-14: sentence too long, to be split in two
Line 17: Please elaborate "*Sediment concentration was estimated both with the observation and simulations, and annual comparisons seemed reasonable for mean annual estimates.*"
What kind of observations, where, how many? Same questions for simulations.
Line 24: what are these BMPs? Not introduced before
Line 27: please elaborate on "*few samples of observed data*"

Other questions/comments arising from the abstract

- Past tense should be used consistently to report on results
- Abstract is lacking of quantitative information on results (values, tests,…)
- Results on sediment concentration and sediment fluxes are missing
- How many runoff and sediment concentration measurements? How? Where? what frequency?
- How many ponds, roads (length, density,..)
- Why using modeling not field data? What distinction between these?
- What about calibration *vs* validation of the model; what about model's sensitivity to roads and dams as compared to other input information?

I hope these comments will be valuable to you.

Tips for improving the structure of the paper:

Abstract

A.      Topic sentence (s) on the subject and research question(s): what is(are) the research gaps in this field of research?

B.      Objectives of the study

C.      Materials and methods used in the study

D.      Main results (with quantitative information, tests of significance)

E.      Conclusions: how these results respond to the objectives; general implications of the research

Introduction sections

A.      Presenting the background of the subject;

B.      Indicating the importance of the research on the subject;

C.      Acknowledging what has be done so far on the subject by referring to existing research studies and reporting ones; referring to methods and ideas associated with other researchers;

D.      Pointing to a gap in knowledge of the subject;

E.      Selecting research objectives

F.      Explaining the organisation of the research;

Discussion section may fulfil one or more of the following functions:

A      Presenting background information

B      Summarising what was (not) done

C      Explaining why it was (not) done

D      Evaluating the method(s) or model used

E      Statement of result(s)

F      Explanation of result(s) – why and how it happened

G      Implication of the result(s) – what it does, or does not, imply

H      Making reference to previous research

I      General statement of interpretation

J      Elaboration of interpretation

K      Discussing implication(s) of the interpretation

L      Rejection of interpretation

M     Acceptance of interpretation

N     Making a recommendation

O     Stating the limitations of the data

P     …………………………….. (other)

Conclusions

A.     Remind of research objectives

B.      Statements of general findings

C.     Statements of specific and significant finding

D.     Statement of overall trends with respect to what was known prior to the study

E.     How well do results respond to initial gaps, research questions

F.     Making predictions; recommendations.

---

## Referee Comment (RC2) · Anonymous Referee #2 · 26 Jan 2016

Manuscript prepared for Hydrol. Earth Syst. Sci.
with version 2015/04/24 7.83 Copernicus papers of the LaTeX class copernicus.cls.
Date: 12 December 2015

[revised manuscript text omitted]

$$Q = \begin{cases} 12,32\,(h[m] - 0,26)^{2,29} & ; \text{para h} < 0,7m \\ 4,27\,h[m] - 1,09 & ; \text{para h} \geqslant 0,7m \end{cases} \tag{4}$$

where $Q$ is streamflow and h is the level of the river.

Some measurements of turbidity and streamflow are also provided (Fig. 13). Based on this data and on Strauch et al. (2013), we described turbidity as function of streamflow, with power ratios and a linear relation (Fig. 14, Eq. 5).

$$TU[NTU] = \begin{cases} 94.582 \cdot Q^{1,1348} & ; \text{para } Q < 0.23 \\ 308.92 \cdot Q^{1,9096} & ; \text{para } 0.23 \leqslant Q < 0.95 \\ 560.42 \cdot Q - 252,3 & ; \text{para } Q \geqslant 0.95 \end{cases} \tag{5}$$

[revised manuscript text omitted]
 Interactions, 14, 1–25, doi:10.1175/2010EI351.1, http://journals.ametsoc.org/doi/abs/10.1175/2010EI351.1, 2010.

Saad, S. I. e. a.: Sensitivity analisys in hydrological and sediment Invest Model and uncertainties of land change cenarios, "in preparation".

SABESP: Relatório Anual de Qualidade da Água, Tech. rep., Governo do Estado de São Paulo, São Paulo, 2014.

Sharp, R., Tallis, H., Ricketts, T., Guerry, A., Wood, S., Chaplin-Kramer, R., Nelson, E., Ennaanay, D., Wolny, S., Olwero, N., Vigerstol, K., Pennington, D., Mendoza, G., Aukema, J., Foster, J., Forrest, J., Cameron, D., Arkema, K., Lonsdorf, E., Kennedy, C., Verutes, G., Kim, C., Guannel, G., Papenfus, M., Toft, J., Marsik, M., Bernhardt, J., Griffin, R., Glowinski, K., Chaumont, N., Perelman, A., Lacayo, M., Mandle, L., Hamel, P., and Vogl, A.: InVEST 3.1.0 Users Guide, The Natural Capital Project, Stanford, http://data.naturalcapitalproject.org/nightly-build/release{_}tip/release{_}tip/InVEST{_}+VERSION+{_}Documentation.pdf, 2014.

Silva, F. d. G. B. d., Minotti, R. T., Neto, F. L., Primavesi, O., and Crestana, S.: Previsão da perda de solo na Fazenda Canchim  SP (EMBRAPA) utilizando geoprocessamento e o USLE 2D, Engenharia Sanitaria e Ambiental, 15, 141–148, http://dx.doi.org/10.1590/S1413-41522010000200006, 2010.

Strauch, M., Lima, J. E. F. W., Volk, M., Lorz, C., and Makeschin, F.: The impact of Best Management Practices on simulated streamflow and sediment load in a Central Brazilian catchment., Journal of environmental management, 127 Suppl, S24–36, doi:10.1016/j.jenvman.2013.01.014, http://www.ncbi.nlm.nih.gov/pubmed/23422359, 2013.

Tallis, H., Ricketts, T., Guerry, A., Wood, S., Sharp, R., Nelson, E., Ennaanay, D., Wolny, S., Olwero, N., Vigerstol, K., Pennington, D., Mendoza, G., Aukema, J., Foster, J., Forrest, J., Cameron, D., Arkema, K., Lonsdorf, E., Kennedy, C., Verutes, G., Kim, C., Guannel, G., Papenfus, M., Toft, J., Marsik, M., and Bernhardt, J.: InVEST 2.4.4 User's Guide, The Natural Capital Project, Stanford, 2011.

Terrado, M., Acuña, V., Ennaanay, D., Tallis, H., and Sabater, S.: Impact of climate extremes on hydrological ecosystem services in a heavily humanized Mediterranean basin, Ecological Indicators, 37, 199–209, doi:10.1016/j.ecolind.2013.01.016, 2014.

Vanmaercke, M., Poesen, J., and Verstraeten, G.: Sediment yield in Europe: spatial patterns and scale dependency, Geomorphology, 130, 142–161, doi:doi:10.1016/j.geomorph.2011.03.010, http://www.sciencedirect.com/science/article/pii/S0169555X11001401, 2011.

White, M. J. and Arnold, J. G.: Development of a simplistic vegetative filter strip model for sediment and nutrient retention at the field scale, Hydrological processes, 23, 1602–1616, doi:10.1002/hyp, 2009.

Yanhe, H. and Chenglong, L.: Advances in the application of the Universal Soil Loss Equation (USLE) in China, Journal of Fujian Agricultural College (Natural Science Edition), 22, 73–77, 1993.

Yuan, Y., Bingner, R. L., and Locke, M. A.: A Review of effectiveness of vegetative buffers on sediment trapping in agricultural areas , Ecohydrology, 2, 321–336, doi:10.1002/eco, 2009.

Zhang, L.: A rational function approach for estimating mean annual evapotranspiration, Water Resources Research, 40, 1–14, doi:10.1029/2003WR002710, 2004.

Zolin, C. a., Folegatti, M. V., Mingoti, R., Paulino, J., Sánchez-Román, R. M., and González, a. M. O.: The first Brazilian municipal initiative of payments for environmental services and its potential for soil conservation, Agricultural Water Management, 137, 75–83, doi:10.1016/j.agwat.2014.02.006, http://dx.doi.org/10.1016/j.agwat.2014.02.006, 2014.

---

## Referee Comment (RC3) · Anonymous Referee #3 · 16 Feb 2016

**Referee comment on the paper:**

**The impact of roads and sediment basins on simulated river discharge and sediment flux in an experimental catchment designed to improve ecosystem services**

By Sandra Isay Saad, Humberto Ribeiro da Rocha, and Jonathan Mota da Silva

**General comments**

After an initial evaluation I consider that this manuscript is not suitable for publication. Not only are the objectives of the paper not very well defined, but the basic data and the technics employed for analysing the results are deficient, ending with a set of rather unsupported scientific conclusions.

In terms of the journal's principal review criteria, it is considered that the manuscript is deficient in terms of scientific significance (it does not represent a substantial contribution in terms of new concepts, ideas, methods or data), scientific quality (approach and methods applied are not conclusive, absence of an appropriate amount of references) and presentation quality (methods, results and conclusions presented in a very unclear and poorly-structured way, inclusion of irrelevant and/or confusing figures, misuse of technical terms, constant errors in spelling and grammar, extremely ambiguous explanations, etc.).

**Specific comments**

In short, the paper attempts to utilize both, a dataset collected in a small catchment in Brazil and an existent hydrological and sediment model, to quantify the annual streamflow and the soil loss during four hydrological years, along with the catchment's soil trapping capacity, including the effects of roads and sediment-traps.

Considering the enormous variability in streamflow but, most importantly, in suspended sediment concentrations in the river (which is usually associated to continuous records of a more easy-to-measure variable like turbidity), the amount and quality of the data presented is poor as to allow an appropriate integration and quantification of the annual water and sediment yields in this small catchment.

On the conceptual side, the authors intend to use data and compare results from very different time scales (from instantaneous or point observations to annual estimations), which compromises the associated calculation uncertainties, and the accuracy and precision of the respective comparisons and conclusions.

Modelling results at the annual scale using methods like the Universal Soil Loss Equation (USLE) are not directly comparable with quantifications from continuous field measurements that could be accounting for process-based events like peak flows, unless special considerations are made. In addition, the authors never mentioned in the manuscript that the USLE provides estimations of sheet erosion only, and that transported erosion (sediment yield) is only a fraction of total erosion (which includes sheet, rill, interrill and gully erosion). Some other methodological deficiencies are

not further commented like. For example, the use of an indirect method to assess the USLE's erosivity factor "R", which certainly compromise the quality of the estimations of annual erosion rates at such small scale. Other difficulties are not mentioned by the authors, like that of utilizing a particular GIS-based routine to assess the length – slope gradient factor (LS) at the catchment scale.

The modelling exercise is not well structured and the objectives are unclear. The flowcharts of calculations from observed and simulated data (Fig. 16) proved that calculations were made in one or another direction, without a clear idea of what its purposes were.

The calibration method seems conceptually unacceptable, since it merely consist in playing with the values of a relatively well defined land use property (the crop factor of pasture), to force the model to produce the desired output.

The conclusions are not relevant, and many of them even acknowledge the significant deficiencies in the data and methodology employed. Sampling and uncertainty is a major problem here, while explicitly expected modelling results could never lead to valid conclusions (i.e., the supposedly proven effects of roads and sediment traps on actual sediment yield in the catchment based only on the results of a model that was explicitly programmed, executed and forced to produce such results). Other unsupported conclusions like that the InVEST model is appropriate "for the mean and current state of the watershed" is not precise and, anyway, not proven according to independent observations or any validation methods accepted in hydrological sciences.

**Technical corrections**

I considered impractical to include technical and typographical corrections to this manuscript, or any typing suggestions, considering they would be too many. I rather encourage the authors to reformulate the paper in both the conceptual and technical aspects, and to fully rewrite it according to a typical structure and style of such a scientific document.

---

## Author Comment (AC1) · 4 Apr 2016

In an urgent need to help preventing environmental resources deterioration, politics involving Payments of Environmental Services have emerged worldwide as a way to recognize financially promoters of improvements on the environmental quality. Conservador das Águas represents the first Brazilian municipal experience of a PES project, and it occurs in a sub-basin of Cantareira's water supply system, that supplies about 50% of water for the Sao Paulo megacity, with 18 million of inhabitants.

Incentivized by the start of the project, since 2009, Posses sub-watershed was feed-up

with some data instrumentation such as pluviometers and rules for measuring the level of the river, which are still few for most environmental studies.

Evaluation of Environmental Services promoted by the project is needed to evaluate the efficiency of some actions taken as reforestation, roads adequation, construction of small basin designed to increase water infiltration nearby the roads„ and to foster new investment in environment.

Studies related to the effects of the land use change on Environmental Services Delivery can be achieved using modelling strategies, which can be performed using good quality observational data, measured for an adequate period, and with high resolution for reasonable spatial representation. This conditions is, however, not easy to be reached, especially in developing countries, for whom such kind of data acquisition may be particular unaffordable.

One of the issues addressed in the paper submitted was the importance of considering the unpaved roads in GIS modeling studies, due to their strong control in sediment export in the watershed, even though they have not been considered in previous GIS modeling studies related to land-use change.

The undoubted efforts performed to calibrate a model intended to predict Environmental Services and its change over land use change, under scarce environmental data conditions was unfortunately misunderstand by reviewers #1 and #3, for whom the paper must be rewritten. Therefore, I kindly ask for the Editor to remove the paper submission so that the main topics will be rewritten, in agreement with the reviewers.

On the other hand, some commentaries were very valuable, especially from reviewer #2, whose suggestions should be certainly used for further publication.

At last, I really appreciate all the effort provided by the Editor, all the HESS team, and reviewers, and I apology for all the inconvenience.

Kind regards, Sandra Saad